# LncRNA *ZNF582-AS1* Expression and Methylation in Breast Cancer and Its Biological and Clinical Implications

**DOI:** 10.3390/cancers14112788

**Published:** 2022-06-04

**Authors:** Junlong Wang, Dionyssios Katsaros, Nicoletta Biglia, Yuanyuan Fu, Chiara Benedetto, Lenora Loo, Zhanwei Wang, Herbert Yu

**Affiliations:** 1University of Hawaii Cancer Center, Honolulu, HI 96813, USA; junlongw@hawaii.edu (J.W.); fuy@hawaii.edu (Y.F.); lloo@cc.hawaii.edu (L.L.); zwang@cc.hawaii.edu (Z.W.); 2Department of Molecular Biosciences & Bioengineering, University of Hawaii at Manoa, Honolulu, HI 96822, USA; 3Department of Surgical Sciences, Gynecology, AOU Città della Salute, University of Torino School of Medicine, 10124 Turin, Italy; dhocc@libero.it (D.K.); chiara.benedetto@unito.it (C.B.); 4Department of Surgical Sciences, Division of Obstetrics and Gynecology, University of Torino School of Medicine, Mauriziano Hospital, 10124 Turin, Italy; nicoletta.biglia@unito.it

**Keywords:** lncRNA, *ZNF582-AS1*, breast cancer, methylation, prognosis

## Abstract

**Simple Summary:**

*ZNF582-AS1* expression is lower in breast cancer compared to adjacent normal tissues, and low expression is associated with poor disease-free and overall survival. Bioinformatic interrogation of *ZNF582-AS1* expression and methylation signatures suggests the lncRNA’s involvement in cell cycle and cell death regulation. The lncRNA may act as a miR-940 sponge, and miR-940 is known to be an onco-miRNA suppressing PTEN. Elevation of transcription factor HIF-1 in cancer cells may repress the expression of *ZNF582-AS1*.

**Abstract:**

*Background:* Long non-coding RNAs (lncRNAs) play an important role in cellular activities and functions, but our understanding of their involvement in cancer is limited. *Methods:* TCGA data on RNA expression and DNA methylation were analyzed for lncRNAs’ association with breast cancer survival, using the Cox proportional hazard regression model. Fresh tumor samples and clinical information from 361 breast cancer patients in our study were used to confirm the TCGA finding on *ZNF582-AS1*. A RT-qPCR method was developed to measure *ZNF582-AS1* expression. Survival associations with *ZNF582-AS1* were verified with a meta-analysis. *In silico* predictions of molecular targets and cellular functions of *ZNF582-AS1* were performed based on its molecular signatures and nucleotide sequences. *Results:*
*ZNF582-AS1* expression was lower in breast tumors than adjacent normal tissues. Low *ZNF582-AS1* was associated with high-grade or ER-negative tumors. Patients with high *ZNF582-AS1* had a lower risk of relapse and death. These survival associations were confirmed in a meta-analysis and remained significant after adjustment for tumor grade, disease stage, patient age, and hormone receptor status. Correlation analysis indicated the possible suppression of *ZNF582-AS1* expression by promoter methylation. Bioinformatics interrogation of molecular signatures suggested that *ZNF582-AS1* could suppress tumor cell proliferation via downregulating the HER2-mediated signaling pathway. Analysis of online data also suggested that HIF-1-related transcription factors could suppress *ZNF582-AS1* expression, and the lncRNA might bind to hsa-miR-940, a known oncogenic miRNA in breast cancer. *Conclusions: ZNF582-AS1* may play a role in suppressing breast cancer progression. Elucidating the lncRNA’s function and regulation may improve our understanding of the disease.

## 1. Introduction

Breast cancer is a common female malignancy worldwide [1]. In the US, more than 281,000 women were diagnosed with breast cancer, and over 43,000 patients died from the disease in 2021 [2]. Although extensive research has revealed many important features and risk factors, our understanding of the malignancy remains limited with respect to molecular changes and biological mechanisms. More research is needed to improve our ability to precisely predict the disease outcomes and effectively treat tumor progression for better survival.

Proteins-coding genes and their products have been a major focus of breast cancer research, addressing the characteristics and behaviors of malignant cells, as well as their responses to treatment. However, investigating proteins is insufficient in understanding the disease from a genomic perspective because protein-coding genes account only for 2% of the genome [3]. Most of the transcripts in human genome (>85%) do not code proteins and are categorized as non-coding RNAs (ncRNAs) [4]. Many ncRNAs are involved in important biological functions and cellular activities, such as transcriptional regulation, genome stability, RNA splicing, and DNA replication [5]. A large number of ncRNAs are long non-coding RNAs (lncRNAs) defined as transcripts with 200 nucleotides or longer [6]. LncRNAs regulate post-transcriptional modifications of mRNA and play crucial roles in epigenetic regulation. Dysregulation of lncRNA activities are found frequently in cancer, and an analysis of their expression may help predict disease outcome and treatment response [7].

The ENCODE project (Encyclopedia of DNA Elements) has identified more than 28,000 lncRNAs in the human genome, but many of them are not well annotated or sufficiently characterized regarding their function and regulation in normal tissue, as well as their involvement in human diseases such as cancer [8]. To identify lncRNAs which may play a role in breast cancer and serve as biomarkers for tumor characterization and disease management, we analyzed a TCGA database on lncRNAs in search for transcripts associated with the disease outcomes. The results of our search were further validated in additional datasets, including a study of our own. In this report, we present our finding on lncRNA *ZNF582-AS1* with respect to its association with breast cancer survival, as well as its possible function and regulation involving tumor progression.

## 2. Materials and Methods

### 2.1. Breast Cancer Patients and Tumor Samples in Our Study

We collected breast tumor samples from 361 patients diagnosed with primary breast cancer, including 239 from the University Hospital between January 1998 and July 1999 and 122 from Mauriziano Hospital between October 1996 and August 2012, at the University of Torino in Italy. Informed consent was obtained from these patients for research use of clinical information and tumor specimens. Fresh tumor samples were collected from the patients during surgery. The tissue specimens were snap-frozen in liquid nitrogen immediately after resection and stored in a −80 °C freezer until analysis. The average age of patients at surgery was 58 years old, and the age range was between 23 and 89 years. Information on disease stage, tumor grade, hormone receptor status, follow-up time, and survival outcome was collected from patient medical records and follow-up visits. The study was approved by the ethic review committees at the University Hospital and Mauriziano Hospital.

### 2.2. RNA Extraction and ZNF582-AS1 Measurement

Total RNAs were extracted from fresh frozen tumor samples (~30 mg), using the Allprep DNA/RNA kit (Qiagen, Germantown, MD, USA), and the RNA samples were treated with RNase-free DNase to remove DNA contamination. The RNA quality was evaluated with NanoDrop (Thermo Fisher Scientific, Waltham, MA, USA). A total of 1 μg of total RNA was reverse-transcribed into cDNA, using the High-Capacity cDNA Reverse Transcription kit (Applied Biosystems, Foster City, CA, USA). The cDNA was used to measure *ZNF582-AS1* expression, and the measurement was performed in LightCycler 480 (Roche, Indianapolis, IN, USA). A specific region of *ZNF582-AS1* was amplified through polymerase chain reaction (PCR), using a commercial PCR kit (LifeTech, Selangor, Malaysia). PCR primers synthesized by IDT (Integrated DNA Technologies, Coralville, IA, USA) were designed to target a common region of *ZNF582-AS1* shared by 3 variant transcripts (NR_037160.1, NR_037161.1, and NR_037159.1). β-actin was used as a control. The primer sequences for *ZNF582-AS1* were 5′-AACGCACGCATCCACTTA-3′ (forward) and 5′-GGTTCCTGGGCTGATTGATAA-3′ (reverse). The β-actin primers were 5′-CACTCTTCCAGCCTTCCTTC-3′ (forward) and 5′-GTACAGGTCTTTGCGGATGT-3′ (reverse). *ZNF582-AS1* expression was calculated as an expression index (EI), using the following formula: 1000 × 2^(−ΔCt)^, where ΔCt = Ct (*ZNF582-AS1*) − Ct (β-actin). Each sample was tested in triplicate, and the replicated results were repeated when coefficients of variation were >15%.

### 2.3. The Cancer Genome Atlas (TCGA) Data

Several TCGA datasets were used in data analysis, including microarray on DNA methylomes and RNA-seq on transcriptomes of mRNAs, lncRNAs, and microRNAs (miRNAs), as well as clinical and follow-up information. RNA-seq data on lncRNAs were downloaded from TANRIC (https://ibl.mdanderson.org/tanric/_design/basic/main.html/ (accessed on 26 November 2019)) [8], which contained expression data on 12,727 lncRNAs in 837 breast cancer samples and 105 adjacent normal breast tissues. RNA-seq data on mRNA expression, as well as clinicopathological data and survival information on overall survival (OS) and disease-free survival (DFS) from the same patients, were downloaded from cBioportal (http://www.cbioportal.org/ (accessed on 26 November 2019)) [9]. Additionally, RNA-seq data on miRNA expression in these patients were obtained from the GDC (Genomic Data Commons) data portal (https://portal.gdc.cancer.gov/ (accessed on 15 June 2020)) [10]. These data were merged together by patient ID.

DNA methylation data from the same patients were downloaded from Wanderer (http://maplab.imppc.org/wanderer/index.html (accessed on 7 May 2020)) [11]. The data included 743 breast cancer samples and 13 CpG sites in the *ZNF582-AS1* promoter (cg01772700, cg24733179, cg11740878, cg09568464, cg02763101, cg22647407, cg08464824, cg13916740, cg24039631, cg20984085, cg25267765, cg07135042, and cg07778983). The methylation data were generated from the Illumina 450K methylation microarray chip and listed in TCGA as level 3 data, which can be linked by patient ID to the transcriptome data from cBioPortal. The merged data on mRNA and methylation contained 463 breast tumor samples, and each sample had mRNA expression on 15,437 genes.

### 2.4. Gene Expression Omnibus (GEO) Data

Nine microarray datasets on gene expression, namely GSE1456 [12], GSE16446 [13], GSE19615 [14], GSE20685 [15], GSE21653 [16], GSE31448 [16], GSE42568 [17], GSE4922 [18], and GSE88770 [19], were downloaded from the Gene Expression Omnibus (GEO) database at NCBI (http://www.ncbi.nlm.nih.gov/geo/ (accessed on 23 July 2020)) [20]. These datasets were selected following the criteria of (a) gene expression in breast cancer samples, (b) information on overall or disease-free survival available, and (c) more than 100 patients in each dataset. For data analysis, we used the expression level from probe 231260_at, which targets *ZNF582-AS1*. Information on each dataset is shown in Appendix A.

### 2.5. Meta-Analysis of ZNF582-AS1 Expression in Association with Survival

In meta-analysis, we classified *ZNF582-AS1* expression into two groups, “*ZNF582-AS1*_High” and “*ZNF582-AS1*_Low”, using the study-specific median as cutoff. Hazards ratio (HR) and 95% confidence interval (CI) were calculated for each dataset, using the Cox proportional hazards regression model. Summary HR and 95% CI were estimated by using the inverse variance weighted method. Since different methods were used for measuring *ZNF582-AS1* expression, we employed the random-effects model to obtain summary results. Review Manager Revman v5.3 (The Nordic Cochrane Centre, Kopenhagen, Denmark) was used for meta-analysis, and the results were presented in Forest Plot. Cochran χ^2^ test and I^2^ statistics were calculated to assess study heterogeneity.

### 2.6. In Silico Prediction of ZNF582-AS1 Function and Regulation

Ingenuity Pathway Analysis (IPA) [21] was used for *in silico* prediction of molecular network and cellular function related to *ZNF582-AS1* expression. The merged expression data on *ZNF582-AS1* and other transcripts included 828 tissue samples and 16,974 mRNAs. Using the median expression as the cutoff, we classified the samples into two groups, *ZNF582-AS1* expression high and low. The mRNA expression in 840 genes was significantly different between the two groups (*p* < 0.05 after the Benjamini–Hochberg adjustment and absolute fold-change greater than 1.5), and these genes were selected as the *ZNF582-AS1* expression signature for IPA interrogation. The names of these genes identified as the *ZNF582-AS1* expression signature are shown in Appendix A.

A similar approach was developed for a promoter methylation signature of *ZNF582-AS1*. DNA methylation data on *ZNF582-AS1* promoter were merged with the matched transcriptomic data. Methylation values in the 12 CpG sites that correlated with *ZNF582-AS1* expression were added together to generate a methylation index for each sample. The median index was used as a cutoff to classify the tumor samples into methylation high and low groups. Genes (*n* = 171) whose mRNA expression was significantly different by the median methylation index (*p* < 0.05) after the Benjamini–Hochberg adjustment and the absolute fold-change higher than 1.2 were selected as the *ZNF582-AS1* promoter methylation signature. The names of the genes identified as the *ZNF582-AS1* methylation signature are shown in Appendix A. The methylation signature was interrogated with IPA and compared with the expression signature to identify complementary signals and functions, as the methylation signature is considered to be the downregulation of *ZNF582-AS1* expression, and the expression signature is deemed to be the upregulation.

To predict the transcription factors of *ZNF582-AS1*, we utilized an online database ChIPBase v2.0 (Guangzhou, China) (http://rna.sysu.edu.cn/chipbase/ (accessed on 16 December 2021)) to interrogate the promoter region of *ZNF582-AS1* between −1 kb upstream to and +1 kb downstream of TSS. This database includes experimental data from the sequencing results of immunoprecipitated chromatin fragments (ChIP-seq) [22].

We used an online tool, LncBase v2. (West Bengal, India) (www.microrna.gr/LncBase/ (accessed on 14 Octomber 2021)) [23], to predict the potential miRNA targets on *ZNF582-AS1*. For the miRNAs whose expression was inversely correlated with *ZNF582-AS1* expression and whose sequences were matched to parts of *ZNF582-AS1,* we considered their possible candidacy as *ZNF582-AS1* targets. TargetScan (Boston, MA, USA) (http://www.targetscan.org/ (accessed on 4 November 2021)) [24] was utilized to predict possible mRNA targets for a given miRNA.

### 2.7. Statistical Analysis

In analyzing data from our own study, *ZNF582-AS1* expressions were classified into 3 groups to assess a dose–response relationship. The classification (high, medium, and low) was based on the tertile distribution of *ZNF582-AS1* expression in the 361 samples. Associations of *ZNF582-AS1* expression with clinicopathological variables, including age at surgery, disease stage, tumor grade, and status of estrogen receptor (ER) and progesterone receptor (PR), were analyzed by using the χ^2^ test. Kaplan–Meier survival curve and log-rank test were used to assess differences in overall and disease-free survival by *ZNF582-AS1* expression in 3 groups. Univariate and multivariate Cox proportional hazards regression analyses were performed to evaluate the associations of disease-free and overall survival with *ZNF582-AS1* expression. Similar survival analyses were also performed for *ZNF582-AS1* promoter methylation, using methylation data in each CpG sites, as well as average methylation in all the CpG sites. In the multivariate analysis of *ZNF582-AS1* expression, the Cox regression models were adjusted for age at surgery, tumor grade, disease stage, and hormone receptor status (ER and PR). Overall survival refers to the time interval from surgery to death or last follow-up. Disease-free survival is the time interval from surgery to disease recurrence or last follow-up. R (version 3.0.2) was used for statistical analysis. The *p*-values < 0.05 (two tailed) were considered to be significant.

## 3. Results

### 3.1. ZNF582-AS1 Expression in Breast Cancer

We analyzed lncRNAs in TANRIC for their relationships with breast cancer survival and found significant associations between *ZNF582-AS1* expression and disease-free and overall survival, i.e., higher expression and lower risk of relapse and death (results shown in meta-analysis). We then compared *ZNF582-AS1* expression in breast cancer (*n* = 837) versus adjacent normal tissue (*n* = 105) in TCGA and observed substantial differences, lower *ZNF582-AS1* expression in cancer than in normal tissues (Figure 1A), and similar differences were also seen in paired tissue samples, i.e., tumor and normal tissues from the same patients (Figure 1B).

### 3.2. ZNF582-AS1 Expression and Patient Characteristics

To confirm the finding in TCGA, we measured *ZNF582-AS1* expression in 361 tumor samples from our own study. Table 1 shows the distributions of *ZNF582-AS1* expression by the clinical and pathological characteristics of breast cancer patients in our study. Low *ZNF582-AS1* expression was associated with high-grade tumors (*p* = 5.86 × 1^0 −5^). A higher percentage (44.9%) of Grade 3 tumors had low *ZNF582-AS1* expression, compared to Grade 1 and 2 tumors, where low expression was seen only in 20% and 22.2%, respectively. More patients with ER-negative tumors had low *ZNF582-AS1* expression (47.3%) compared to those with ER-positive tumors (27.8%) (*p* = 0.003). Similar trends were also indicated for PR status (*p* = 0.053) and disease stage (*p* = 0.055), with lower expression in PR-negative tumors or in patients with advanced stage disease. No difference was observed in *ZNF582-AS1* expression by patient age at surgery (*p* = 0.965).

### 3.3. ZNF582-AS1 Expression and Breast Cancer Survival

The Kaplan–Meier survival curves showed that patients with a high *ZNF582-AS1* expression had better disease-free (*p* = 0.001) and overall survival (*p* = 0.011) compared to those with low expression (Figure 2A,B). Patients with high *ZNF582-AS1* had a more than 60% reduction in risks of relapse (HR = 0.34, 95%CI: 0.19–0.61, *p* < 0.001) and death (HR = 0.36, 95%CI: 0.18–0.73, *p* = 0.005) (Table 2). In the multivariate analysis, where patient age at surgery, tumor grade, disease stage, and ER/PR status were adjusted, *ZNF582-AS1* expression remained significantly associated with relapse (HR = 0.42, 95%CI: 0.21–0.61, *p* = 0.012), but not with death (*p* = 0.373).

### 3.4. Meta-Analysis of ZNF582-AS1 in Association with Survival

A meta-analysis was performed to examine the association between *ZNF582-AS1* expression and breast cancer survival in nine GEO datasets, the TCGA data, and our own study. In total, 7 datasets had information on overall survival in 1999 patients, and 10 datasets had information on disease-free survival in 2649 patients. Most of the studies showed a similar direction of associations between breast cancer survival and *ZNF582-AS1* expression (Figure 3A,B). The summary results indicated that both overall and disease-free survival were significantly associated with *ZNF582-AS1* expression. Compared to those with low expression, patients with high expression had a lower risk of death (HR = 0.62, 95%CI: 0.51–0.76) and relapse (HR = 0.71, 95%CI: 0.57–0.89). These results did not change substantially when only GEO datasets were included in the analysis (data not shown).

### 3.5. DNA Methylation and ZNF582-AS1 Expression

Using the methylation and expression data from TCGA, we analyzed the correlation between DNA methylation in 13 CpG sites and *ZNF582-AS1* expression in breast cancer. Our analysis showed that DNA methylation levels in the *ZNF582-AS1* promoter were inversely correlated with its expression in 12 of the 13 CpG sites, i.e., higher methylation and lower expression (Appendix A). This finding suggests that promoter methylation in the *ZNF582-AS1* gene may suppress its expression. We also evaluated the relationship between DNA methylation and breast cancer survival and found no association with either overall or disease-free survival (Appendix A), suggesting that promoter methylation may suppress *ZNF582-AS1* expression, but the expression is regulated by multiple factors, in addition to methylation; the final expression level in the tumor plays a role in survival outcomes.

### 3.6. Bioinformatic Interrogation of ZNF582-AS1 Expression and Methylation Signatures

Using IPA, we interrogated the signatures of *ZNF582-AS1* expression and methylation and found complementary results between the signatures. Both signatures showed similar molecular and cellular functions in four of the top five predictions, namely (1) cell cycle; (2) cellular assembly and organization; (3) DNA replication, recombination, and repair; and (4) cell death and survival (Figure 4A,B). The expression signature suggested the upregulation of ERBB2, AREG, FOXM1, and RABL6 and downregulation of E2F6, whereas methylation signature showed opposite effects on these molecules (Figure 4C,D). Furthermore, the expression signature was linked to the suppression of tumor cell proliferation and colony formation, and promotion of cell apoptosis and senescence, whereas the methylation signature indicated promotion of tumor cell proliferation and viability, and suppression of cell death and senescence (Figure 4C,D).

### 3.7. ZNF582-AS1 and hsa-miR-940

Using LncBase, we predicted which miRNAs might be targeted by *ZNF582-AS1* among those whose expressions were inversely correlated with *ZNF582-AS1* expression, and seven potential candidates were indicated, namely hsa-miR-130b-5p, hsa-miR-590-5p, hsa-miR-627-3p, hsa-miR-939-3p, hsa-miR-940, hsa-miR-3682-3p, and hsa-miR-4746-5p (Appendix A). The prediction results also showed that four of the miRNAs, hsa-miR-939-3p, hsa-miR-940, hsa-miR-3682-3p, and hsa-miR-4746-5p, might be able to interact with all three variant transcripts of *ZNF582-AS1*. Of them, hsa-miR-940 was known to have possible oncogenic effects on breast cancer. Using TargetScan, we predicted the possible mRNAs targeted by hsa-miR-940 and found that PTEN was one of the possible targets (Appendix A).

To evaluate the involvement of hsa-miR-940 in breast cancer, we compared hsa-miR-940 expression between breast cancer and adjacent normal tissues in 105 TCGA samples (Figure 5A) and found higher expression in breast cancer than in normal tissues (7.09 vs. 1.40; *p* = 3.05 × 10 ^−10^). The relationship between hsa-miR-940 expression and overall survival was analyzed in 988 breast cancer patients after hsa-miR-940 expressions were grouped into three categories based on the tertile distribution (high, medium, and low). Patients with a high expression had poor survival when compared to those with a low expression (*p* = 0.0136) (Figure 5B), suggesting that hsa-miR-940 may negatively impact breast cancer survival, an effect opposite to that of *ZNF582-AS1*. A predicted RNA network among lncRNA *ZNF582-AS1*, miRNA has-miR-940, and mRNA PTEN is shown in Figure 6.

### 3.8. Regulation of ZNF582-AS1 Expression

We used an online database (ChIPBase v2.0) to predict the transcription factors for *ZNF582-AS1*, *chr19:56*, *393*, *312-56*, and *414*, *800 (GRCh38/hg38)*, and the online tool suggested five potential candidates in breast cancer cell lines, namely CTCF, KDM5B, HIF1A, ARNT, and NRF1 (Appendix A). The transcriptional start site (TSS) of *ZNF582-AS1* is located at chr19:56393655. The binding sites predicted for HIF1A, ARNT, and NRF1 were in the regions of chr19:56393150-56393909, chr19:56393174-56393888, and chr19:56393152-56393920, respectively, all of which included TSS, suggesting that their binding to *ZNF582-AS1* may suppress its expression.

## 4. Discussion

In the study, we searched for lncRNAs in TCGA for their possible involvement in breast cancer and found substantial differences in *ZNF582-AS1* expression between breast cancer and adjacent normal tissues, lower in tumors compared to normal tissues. The analysis of TCGA data further indicated that low expression was correlated with promoter methylation, and *ZNF582-AS1* expression in breast cancer was associated with tumor grade and hormone receptor status. In addition, patients with a high *ZNF582-AS1* expression had favorable disease-free and overall survival. These associations were independent from clinical and pathological factors and were confirmed in the meta-analysis of multiple independent studies, including 1999 patients with overall survival and 2649 patients with disease-free survival. Our finding suggests that *ZNF582-AS1* may be a possible tumor suppressor in breast cancer with a potential to serve as a biomarker for breast cancer prognosis. 

In addition to breast cancer, low *ZNF582-AS1* was also found in several other cancers. Kumegawa et al. reported that *ZNF582-AS1* was significantly lower in colorectal cancer than in normal tissues, and *ZNF582-AS1* expression was inversely related to aberrant DNA methylation [25]. Cheng et al., found low *ZNF582-AS1* in renal cell carcinoma [26], and Yuan et al., observed lower *ZNF582-AS1* expression in cervical cancer than in adjacent normal tissues [27]. A recent publication indicated that *ZNF582-AS1* expression was downregulated in clear cell kidney cancer due to DNA hypermethylation, and downregulation promoted tumor growth and metastasis [28]. Despite the consistent findings of low *ZNF582-AS1* in multiple cancer sites and low expression associated with tumor progression, our understanding of the molecular mechanisms underlying the involvement of *ZNF582-AS1* in cancer remains limited.

*ZNF582-AS1* is also referred to as *ZNF582-DT* (ZNF582 divergent transcript). The gene encoding *ZNF582-AS1* is in chromosome 19 (19q13.43) with 5520 nucleotide bases in length. The *ZNF582-AS1* gene is adjacent to the *ZNF582* gene in a head-to-head orientation, sharing a common promoter region, but their transcriptions are in opposite directions. Furthermore, the two genes are totally different in DNA sequences when using NCBI BLAST to compare. Several studies reported that *ZNF582* expression was low in cancer, and low expression was associated with promoter hypermethylation and disease outcomes [29,30,31,32,33,34]. To assess if the two genes have any relationship in expression, we analyzed their correlation in breast cancer. The analysis showed that levels of *ZNF582-AS1* and *ZNF582* expression were positively correlated (r = 0.754; *p* = 1.32 × 10^−154^), suggesting a possible coordination between these genes in expression. As *ZNF582* expression is regulated by promoter methylation, we also analyzed *ZNF582-AS1* promoter methylation in relation to its expression and found an indication of similar regulation. However, we found no association between *ZNF582* expression and survival outcomes, suggesting that the two genes are different in function, although they may share certain similarities in expression regulation.

To explore the signal pathways and cellular functions of *ZNF582-AS1*, we developed two molecular signatures of *ZNF582-AS1* in breast cancer, one based on its transcriptome and another involving its methylome, which were considered to represent high and low expression of *ZNF582-AS1*, respectively, and to presumably be complementary to one another in cell signaling. Interestingly, our bioinformatic interrogation did identify similar signal pathways in opposite directions of activity; that is, high expression led to the downregulation of ERBB2, AREG, FOXM1, and RABL6, all of which were in an opposite direction of regulation when the methylation signature (low expression) was interrogated. These bioinformatic results were obtained independently based on two signatures established separately from the transcriptomes and methylomes of *ZNF582-AS1*. The key proteins downregulated by *ZNF582-AS1* are known products of oncogenes or proto-oncogenes. ERBB2 is HER2, a member of the epidermal growth factor receptor (EGFR) family [35] which is often overexpressed in breast cancer and an effective drug target in treating HER2-positive breast cancer [36]. AREG (amphiregulin), a member of the EGF family and a ligand for EGFR or TNF-alpha receptor [37], is involved in the initiation and progression of breast cancer [38]. High FOXM1 expression has been found in several types of cancer, including breast cancer [39], and the protein involves transcription activation and regulation of cell proliferation [40,41]. FOXM1 is positively correlated with HER2 in breast cancer [42]. RABL6 is a member of the RAS oncogene family, and its expression is elevated in breast cancer cell lines and tumor samples. Suppressing RABL6 expression inhibits tumor cell proliferation and invasion, while low RABL6 increases cell-cycle arrest and apoptosis. RABL6 overexpression also indicates poor prognosis of breast cancer [43].

One of the lncRNA’s functions is believed to be achieved through its complementary binding to other RNA transcripts [44], which can happen either in physiological or pathological conditions [45,46]. In search for potential binding targets of *ZNF582-AS1*, we analyzed the correlation of *ZNF582-AS1* expression with hundreds of miRNAs in TCGA. Through the analysis, we found a number of miRNAs whose expression were correlated with *ZNF582-AS1* expression. Using LncBase’s Prediction Module, we identified several miRNAs with the potential to bind to *ZNF582-AS1*. Combining the two lists, we found seven miRNAs which may bind to *ZNF582-AS1* and are also inversely correlated with its expression. Among them, hsa-miR-940 may interact with all three variant transcripts of *ZNF582-AS1*. A recent study showed that hsa-miR-940 could increase the proliferation and invasion of breast cancer cells [47]. We also found that hsa-miR-940 was significantly higher in breast tumors than in adjacent normal tissues and that high hsa-miR-940 was associated with poor breast cancer survival. TargetScan predicted that hsa-miR-940 could interact with the 3′UTR of PTEN mRNA. All of this suggests that the effect of *ZNF582-AS1* on breast cancer may be achieved through its binding to hsa-miR-940, resulting in the miRNA’s inability to suppress PTEN translation in breast cancer.

Our ChIPBase search for transcription factors of *ZNF582-AS1* indicated that HIF1A, ARNT, and NRF1 might be potential candidates for suppressing *ZNF582-AS1* transcription. HIF1A is a subunit of HIF-1 (hypoxia-inducible factor 1), and ARNT (aryl hydrocarbon receptor nuclear translocator) is a co-factor for transcriptional regulation by HIF-1. HIF-1 overexpression is implicated in tumor growth [48]. Nuclear respiratory factor 1 (NRF1) is a transcription factor regulating important metabolic genes involved in cell growth and is found to behave similar to an oncoprotein in breast cancer [49]. Based on their predicted binding motifs encompassing the transcriptional start site of *ZNF582-AS1*, we speculate that *ZNF582-AS1* expression may be downregulated in breast cancer by these transcription factors [50,51,52].

In summary, our study showed that *ZNF582-AS1* was low in breast cancer, and low *ZNF582-AS1* was associated with aggressive breast cancer. Patients with high *ZNF582-AS1* had better overall and disease-free survival compared to those with low *ZNF582-AS1*. These survival associations were confirmed in the meta-analysis of multiple datasets. Promoter methylation may inhibit *ZNF582-AS1* expression, and hypoxia-related transcription factors may also downregulate its transcription. Bioinformatic interrogation of *ZNF582-AS1* signatures in transcriptome and methylome suggests that *ZNF582-AS1* may suppress tumor cell proliferation and viability and promote apoptosis via inhibition of the HER2-related signaling pathway. The *in silico* analysis also indicates that *ZNF582-AS1* may bind to hsa-miR-940, which is known to be an onco-miRNA possibly suppressing PTEN. If our findings are confirmed in additional studies, this lncRNA may serve as a useful biomarker for breast cancer prognosis and treatment.

## Figures and Tables

**Figure 1 cancers-14-02788-f001:**
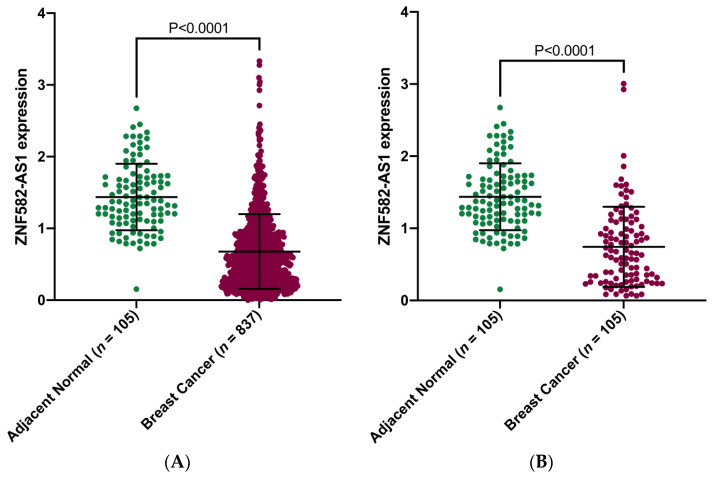
Comparison of *ZNF582-AS1* expression between breast cancer and adjacent normal tissues. (**A**) *ZNF582-AS1* expression in breast cancer (*n* = 837) and adjacent breast tissues (*n* = 105). (**B**) *ZNF582-AS1* expression in matched breast cancer (*n* = 105) and adjacent breast tissues (*n* = 105).

**Figure 2 cancers-14-02788-f002:**
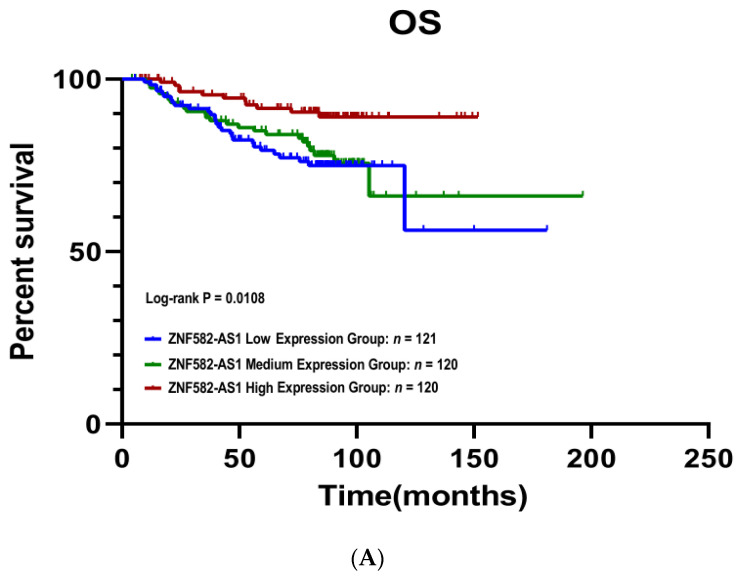
Kaplan–Meier survival curves by *ZNF582-AS1* expression in tertile. (**A**) Overall survival (OS) curves by high, mid, and low expression of *ZNF582-AS1*. (**B**) Disease-free survival (DFS) curves by high, mid, and low expression of *ZNF582-AS1*.

**Figure 3 cancers-14-02788-f003:**
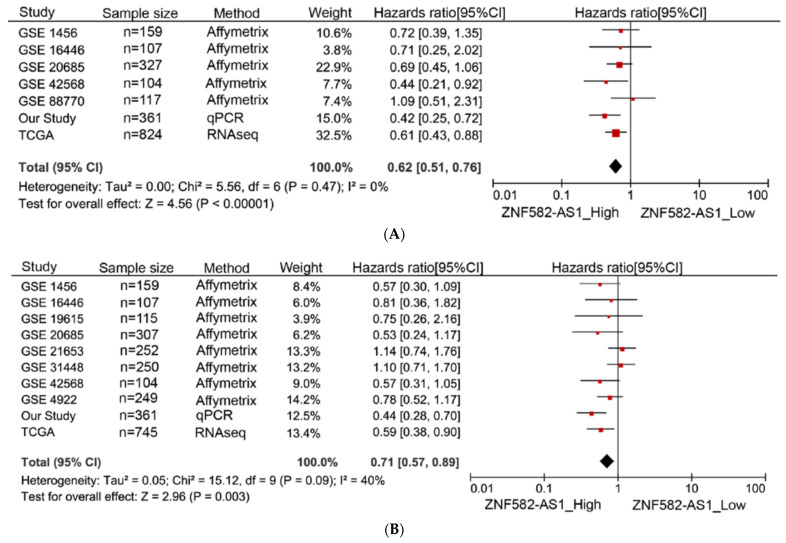
Meta-analysis of associations between *ZNF582-AS1* expression and breast cancer survival. (**A**) *ZNF582-AS1* expression (high vs. low) in association with overall survival (OS). (**B**) *ZNF582-AS1* expression (high vs. low) in association with disease-free survival (DFS).

**Figure 4 cancers-14-02788-f004:**
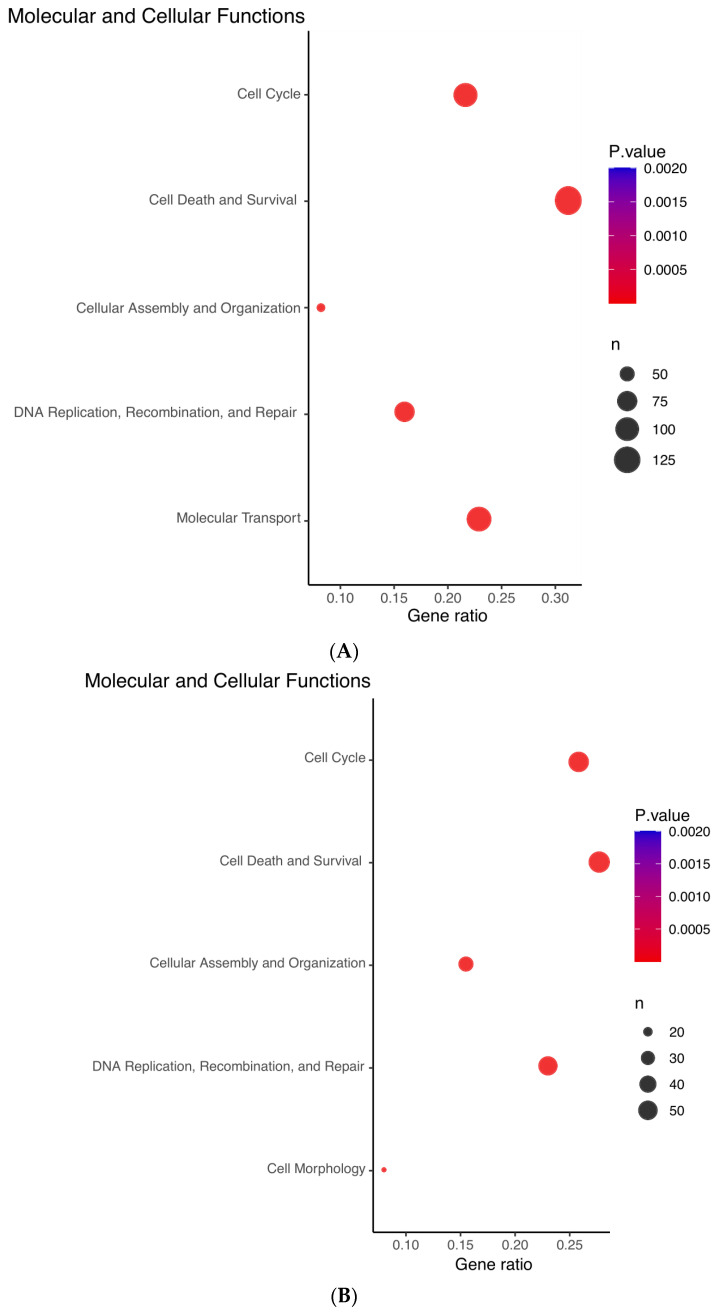
Ingenuity Pathway Analysis (IPA) prediction of the molecular and cellular functions of *ZNF582-AS1* based on its expression and methylation signatures. (**A**) IPA prediction of the molecular and cellular functions related to the *ZNF582-AS1* expression signature. (**B**) IPA prediction of the molecular and cellular functions related to the *ZNF582-AS1* promoter methylation signature. (**C**) IPA prediction of the signal network related to the *ZNF582-AS1* expression signature. (**D**) IPA prediction of the signal network related to the *ZNF582-AS1* promoter methylation signature.

**Figure 5 cancers-14-02788-f005:**
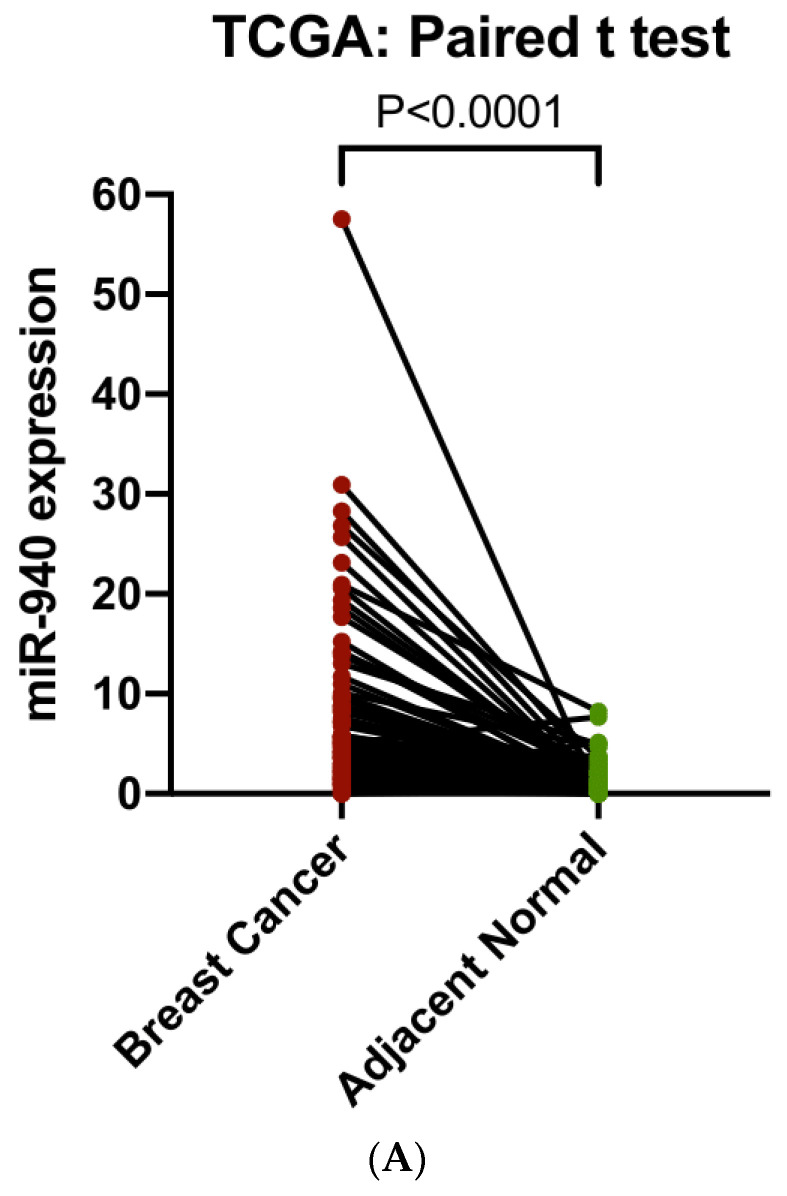
Analysis of hsa-miR-940 expression in breast cancer and adjacent normal breast tissues (*n* = 105), as well as its expression in association with overall survival in TCGA. (**A**) Comparison of hsa-miR-940 expression between breast cancer and match adjacent normal tissues (*n* = 105). (**B**) Kaplan–Meier overall survival (OS) curves by high, mid, and low expression of hsa-miR-940.

**Figure 6 cancers-14-02788-f006:**
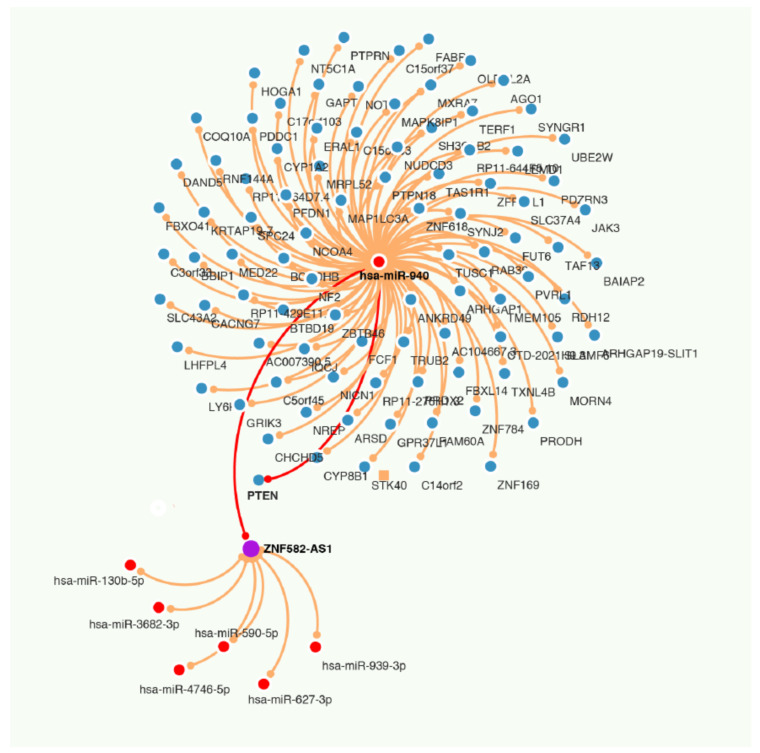
RNA network among lncRNA *ZNF582-AS1*, miRNA has-miR-940, and mRNA PTEN.

**Table 1 cancers-14-02788-t001:** Associations between *ZNF582-AS1* expression and clinicopathological characteristics of breast cancer.

		*ZNF582-AS1* Expression			
Variables	Total (*n* = 361) No. (%)	Low No. (%)	Mid No. (%)	High No. (%)	*p*-Value
Age					
Age < 58.14	181 (50.14)	61 (33.70)	59 (32.60)	61 (33.70)	0.965
Age ≥ 58.14	180 (49.86)	60 (33.33)	61 (33.89)	59 (32.78)	
Disease stage					
Stage I	114 (33.33)	28 (24.56)	39 (34.21)	47 (41.23)	0.055
Stage II	174 (50.88)	59 (33.91)	59 (33.91)	56 (32.18)	
Stage III and IV	54 (15.79)	27 (50.00)	16 (29.63)	11 (20.37)	
Tumor grade					
Grade 1	40 (11.33)	8 (20.00)	11 (27.50)	21 (52.50)	5.86 × 10^−5^
Grade 2	135 (38.24)	30 (22.22)	47 (34.81)	58 (42.96)	
Grade 3	178 (50.42)	80 (44.94)	58 (32.58)	40 (22.47)	
ER status					
Positive	245 (68.63)	68 (27.76)	85 (34.69)	92 (37.55)	0.003
Negative	112 (31.37)	53 (47.32)	32 (28.57)	27 (24.11)	
PR status					
Positive	212 (59.38)	60 (28.30)	73 (34.43)	79 (37.26)	0.053
Negative	145 (40.62)	61 (42.07)	44 (30.34)	40 (27.59)	

**Table 2 cancers-14-02788-t002:** Associations between *ZNF582-AS1* expression and risk of breast cancer relapse or death.

*ZNF582-AS1* Expression		Relapse			Death	
	HR	95%CI	*p*-Value	HR	95%CI	*p*-Value
Univariate analysis						
Low	1			1		
Mid	0.70	0.43–1.13	0.143	0.88	0.51–1.51	0.646
High	0.34	0.19–0.61	<0.001	0.36	0.18–0.73	0.005
Multivariate analysis *						
Low	1			1		
Mid	0.84	0.49–1.42	0.507	1.15	0.62–2.14	0.646
High	0.42	0.21–0.61	0.012	0.71	0.33–1.52	0.373

* Adjusted for age at surgery, disease stage, tumor grade, and ER and PR status.

## Data Availability

Access to public data is described in the manuscript. Information on our study is available upon request.

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
