# Peer review of "LncRNA ZNF582-AS1 Expression and Methylation in Breast Cancer and Its Biological and Clinical Implications"

_cancers, 2022, doi:10.3390/cancers14112788_

Round 1

Reviewer 1 Report

Reviewer Comments:

Junlong Wang and coworkers present the manuscript entitled “LncRNA ZNF582-AS1 expression and methylation in breast cancer and its biological and clinical implications”. This is another TGCA-based paper, which in general are very preliminary. A major concern must be addressed before potential publication of this manuscript in Cancers journal.

Major comments

  1. The study us very preliminary “expression“ study and to be complete must include functional analysis of LncRNA ZNF582-AS1

Minor comments:

  1. Authors should include the graphical abstract and simple summary.
  2. In the introduction as background, the authors should include what is known about lncRNA ZNF582-AS1 in breast cancer and other types of cancer. For example, the article Yang, W., et al (2021). Downregulation of lncRNA ZNF582-AS1 due to DNA hypermethylation promotes clear cell renal cell carcinoma growth and metastasis by regulating the N(6)-methyladenosine modification of MT-RNR1. Journal of experimental & clinical cancer research : CR40(1), 92. https://doi.org/10.1186/s13046-021-01889-8
  3. Please, Indicate panel A in Figure 1.
  4. In figure 1 legend, delete "AA"
  5. Page 11, delete figure 1B
  6. Figure 4. Authors should show molecular and cellular functions in a bubble chart.
  7. The authors should include a network of the ZNF582-AS1 miRNA targets.

Author Response

Reviewer 1

Major comments

  1. The study us very preliminary “expression“ study and to be complete must include functional analysis of LncRNA ZNF582-AS1

Response:

We agree with the reviewer that functional analysis is important for us to find evidence to support the clinical association with expression.  However, we have limited laboratory capacity to perform in vitro or in vivo experiments.  To address this limitation, we tool an alternative approach by conducting bioinformatic analyses on two large omics datasets and using complementary computational tools and databases.  Our evaluation on potential function is based on two ZNF582-AS1-associated molecular signatures: expression and methylation, which mimics the knockdown and knock-in cell experiments.  The signatures consist of the genes which show substantial differences in expression and methylation when comparing high versus low ZNF582-AS1 expression.  These molecular signatures are complementary to the lncRNA phenotype, i.e., high expression corresponding to low methylation.  The signatures were interrogated with IPA (Ingenuity Pathway Analysis), a widely used bioinformatic method, to predict biological functions and signal network. The predicted functions are surprisingly identical, including control of cell cycle, regulation of cell death, and DNA repair, which appear to be consistent with the finding of our clinical studies where high expression is associated with favorable survival outcomes.  More importantly, these signatures point to the same functions from different angles, i.e., gene expression versus DNA methylation, suggesting that the prediction is quite reliable.

Minor comments:

  1. Authors should include the graphical abstract and simple summary.

Response:

Following the suggestion, we have included a graphical abstract and simple summary in the revised manuscript.

  1. In the introduction as background, the authors should include what is known about lncRNA ZNF582-AS1 in breast cancer and other types of cancer. For example, the article Yang, W., et al (2021). Downregulation of lncRNA ZNF582-AS1 due to DNA hypermethylation promotes clear cell renal cell carcinoma growth and metastasis by regulating the N(6)-methyladenosine modification of MT-RNR1. Journal of experimental & clinical cancer research : CR40(1), 92. https://doi.org/10.1186/s13046-021-01889-8

Response:

The suggested information is provided in the second paragraph of Discussion, including research findings of ZNF582-AS1 in different cancers by other investigators and the reference mentioned by the reviewer. The reason for us to include the information in Discussion, not in Introduction, is that ZNF582-AS1 is the finding of our study, not the aim of our initial investigation.  That is why we discuss this finding extensively in Discussion to provide more complete information on ZNF582-AS1 as well as its clinical and biological implications.

  1. Please, Indicate panel A in Figure 1. 

Response:

Our original figure file was fine without this problem.  An error occurred during the merge of our manuscript file with the figure since they were uploaded separately, and we did not see the merged document.  We will make sure it does not happen again.

  1. In figure 1 legend, delete "AA".  

Response:

Our original figure file was fine without this problem.  An error occurred during the merge of our manuscript file with the figure since they were uploaded separately, and we did not see the merged document.  We will make sure it does not happen again.

  1. Page 11, delete figure 1B. 

Response:

This error occurred during the merge of our manuscript with the figure file since they were uploaded separately, and we did not see the merged document.  We will make sure it does not happen again.

  1. Figure 4. Authors should show molecular and cellular functions in a bubble chart.

Response:

Following the suggestion, we have changed the tables (Table 4A and 4B) to bubble charts shown below. 

  1. The authors should include a network of the ZNF582-AS1 miRNA targets.figure?

Response:

Based on the comment, we have included a new figure (Figure 6) to show the network among lncRNA, miRNA, and mRNA.

Figure 6

Reviewer 2 Report

Major points:

  1. Line 260-262, “We also evaluated the relationship between DNA methylation and breast cancer survival and found no association with either overall or disease-free survival (data not shown).” Do you mean specifically the DNA methylation levels in the ZNF582-AS1 promoter? Please show the data and elaborate the implications of the results.
  2. Between section 3.5 and 3.6, format of the document is off. Likely an editorial error?
  3. Please discuss further how “the head-to-head orientation between ZNF582-AS1 and ZNF582” and their sequence similarity, or dissimilarity, may or may not interfere with the results of the existing datasets analyzed in this study and your interpretation from the analyses.

Minor points:

  1. Line 53,please explain what could be pre-transcriptional modifications of mRNA?
  2. Fig 1A, a label is cut off.
  3. Line 308, “…and interact with ZNF582-AS1.” This cannot be concluded from the expression of hsa-miR-940 and the patient survival.
  4. Line 322, please state the version of the reference genome.
  5. Line 322-325, please list some reference papers to support the statement.

Author Response

Reviewer 2

Major points:

  1. Line 260-262, “We also evaluated the relationship between DNA methylation and breast cancer survival and found no association with either overall or disease-free survival (data not shown).” Do you mean specifically the DNA methylation levels in the ZNF582-AS1 promoter? Please show the data and elaborate the implications of the results.

Response:

Yes, DNA methylation levels in the lncRNA’s promoter.  These results are now provided in Supplementary Figure S1. We also elaborate this finding in our revision. Please see below and in the manuscript.

Revision:

We also evaluated the relationship between DNA methylation and breast cancer survival and found no association with either overall or disease-free survival (Supplementary Figure S1), suggesting that promoter methylation may suppress ZNF582-AS1 expression, but the expression is regulated by multiple factors in addition to methylation; the final expression level in the tumor plays a role in survival outcomes.

  1. Between section 3.5 and 3.6, format of the document is off. Likely an editorial error?

Response:

This error occurred during the merge of our manuscript file with the figure since they were uploaded separately, and we did not see the merged file.  We will make sure it does not happen again.

  1. Please discuss further how “the head-to-head orientation between ZNF582-AS1 and ZNF582” and their sequence similarity, or dissimilarity, may or may not interfere with the results of the existing datasets analyzed in this study and your interpretation from the analyses.

Response:

The head-to-head orientation between ZNF582-AS1(blue highlighted) and ZNF582 are shown below from the USCS Genome Browser. From the picture, we can see that ZNF582-AS1 and ZNF582 have different transcription directions, and the transcriptional regions do not overlap. The two genes are different in sequences.

We further compared the sequence similarities between the two transcripts using NCBI BLAST.  Our comparison showed no sequence similarity between ZNF582-AS1 and ZNF582.  The result is shown below in orange shade.

We also analyzed ZNF582 expression in relation to survival outcomes and found no associations with either disease-free or overall survival. The Kaplan-Meier survival curves are shown below.

Revision:

The ZNF582-AS1 gene is adjacent to the ZNF582 gene in a head-to-head orientation, sharing a common promoter region, but their transcriptions are in opposite directions.  Further, the two genes are totally different in DNA sequences when using NCBI BLAST to compare.

To assess if the two genes have any relationship in expression, we analyzed their correlation in breast cancer.  The analysis showed that levels of ZNF582-AS1 and ZNF582 expression were positively correlated (r=0.754, p=1.32E-154), suggesting a possible coordination between these genes in expression.  As ZNF582 expression is regulated by promoter methylation, we also analyzed ZNF582-AS1 promoter methylation in relation to its expression and found an indication of similar regulation.  However, we found no association between ZNF582 expression and survival outcomes, suggesting that the two genes are different in function although they may share certain similarities in expression regulation.    

Minor points:

  1. Line 53, please explain what could be pre-transcriptional modifications of mRNA?

Response:

We have revised the sentence by removing ‘pre-transcriptional modification of mRNA’.

Revision:

LncRNAs regulate post-transcriptional modifications of mRNA and play crucial roles in epigenetic regulation. 

  1. Fig 1A, a label is cut off.

Response:

This error occurred during the merge of our manuscript file with the figure since they were uploaded separately, and we did not see the merged file.  We will make sure it does not happen again.

  1. Line 308, “…and interact with ZNF582-AS1.” This cannot be concluded from the expression of hsa-miR-940 and the patient survival.

Response:

We agree with the reviewer and have revised the sentence. Our revision is “Patients with high expression had poor survival comparing to those with low expression (p=0.0136) (Figure 5B), suggesting that high hsa-miR-940 may negatively impact on breast cancer survival, an effect which is opposite to that of ZNF582-AS1.”

  1. Line 322, please state the version of the reference genome.

Response:

The reference genome version has been included as chr19:56,393,312-56,414,800 (GRCh38/hg38)

  1. Line 322-325, please list some reference papers to support the statement.

Response:

The references listed below have been added to the manuscript as reference 50-52. These literatures support our speculation that transcription factors with binding motifs encompassing the transcriptional start site (TSS) tend to have a repressing effect on gene transcription.

  1. Kannan Tharakaraman, Olivier Bodenreider, David Landsman, John L. Spouge, Leonardo Mariño-Ramírez, The biological function of some human transcription factor binding motifs varies with position relative to the transcription start site, Nucleic Acids Research, Volume 36, Issue 8, 1 May 2008, Pages 2777–2786, https://doi.org/10.1093/nar/gkn137
  2. Tabach Y, Brosh R, Buganim Y, Reiner A, Zuk O, Yitzhaky A, et al. (2007) Wide-Scale Analysis of Human Functional Transcription Factor Binding Reveals a Strong Bias towards the Transcription Start Site. PLoS ONE 2(8): e807. https://doi.org/10.1371/journal.pone.0000807
  3. Whitfield, T.W., Wang, J., Collins, P.J. et al. Functional analysis of transcription factor binding sites in human promoters. Genome Biol 13, R50 (2012). https://doi.org/10.1186/gb-2012-13-9-r50

Reviewer 2

Major points:

  1. Line 260-262, “We also evaluated the relationship between DNA methylation and breast cancer survival and found no association with either overall or disease-free survival (data not shown).” Do you mean specifically the DNA methylation levels in the ZNF582-AS1 promoter? Please show the data and elaborate the implications of the results.

Response:

Yes, DNA methylation levels in the lncRNA’s promoter.  These results are now provided in Supplementary Figure S1. We also elaborate this finding in our revision. Please see below and in the manuscript.

Revision:

We also evaluated the relationship between DNA methylation and breast cancer survival and found no association with either overall or disease-free survival (Supplementary Figure S1), suggesting that promoter methylation may suppress ZNF582-AS1 expression, but the expression is regulated by multiple factors in addition to methylation; the final expression level in the tumor plays a role in survival outcomes.

  1. Between section 3.5 and 3.6, format of the document is off. Likely an editorial error?

Response:

This error occurred during the merge of our manuscript file with the figure since they were uploaded separately, and we did not see the merged file.  We will make sure it does not happen again.

  1. Please discuss further how “the head-to-head orientation between ZNF582-AS1 and ZNF582” and their sequence similarity, or dissimilarity, may or may not interfere with the results of the existing datasets analyzed in this study and your interpretation from the analyses.

Response:

The head-to-head orientation between ZNF582-AS1(blue highlighted) and ZNF582 are shown below from the USCS Genome Browser. From the picture, we can see that ZNF582-AS1 and ZNF582 have different transcription directions, and the transcriptional regions do not overlap. The two genes are different in sequences.

We further compared the sequence similarities between the two transcripts using NCBI BLAST.  Our comparison showed no sequence similarity between ZNF582-AS1 and ZNF582.  The result is shown below in orange shade.

We also analyzed ZNF582 expression in relation to survival outcomes and found no associations with either disease-free or overall survival. The Kaplan-Meier survival curves are shown below.

Revision:

The ZNF582-AS1 gene is adjacent to the ZNF582 gene in a head-to-head orientation, sharing a common promoter region, but their transcriptions are in opposite directions.  Further, the two genes are totally different in DNA sequences when using NCBI BLAST to compare.

To assess if the two genes have any relationship in expression, we analyzed their correlation in breast cancer.  The analysis showed that levels of ZNF582-AS1 and ZNF582 expression were positively correlated (r=0.754, p=1.32E-154), suggesting a possible coordination between these genes in expression.  As ZNF582 expression is regulated by promoter methylation, we also analyzed ZNF582-AS1 promoter methylation in relation to its expression and found an indication of similar regulation.  However, we found no association between ZNF582 expression and survival outcomes, suggesting that the two genes are different in function although they may share certain similarities in expression regulation.    

Minor points:

  1. Line 53, please explain what could be pre-transcriptional modifications of mRNA?

Response:

We have revised the sentence by removing ‘pre-transcriptional modification of mRNA’.

Revision:

LncRNAs regulate post-transcriptional modifications of mRNA and play crucial roles in epigenetic regulation. 

  1. Fig 1A, a label is cut off.

Response:

This error occurred during the merge of our manuscript file with the figure since they were uploaded separately, and we did not see the merged file.  We will make sure it does not happen again.

  1. Line 308, “…and interact with ZNF582-AS1.” This cannot be concluded from the expression of hsa-miR-940 and the patient survival.

Response:

We agree with the reviewer and have revised the sentence. Our revision is “Patients with high expression had poor survival comparing to those with low expression (p=0.0136) (Figure 5B), suggesting that high hsa-miR-940 may negatively impact on breast cancer survival, an effect which is opposite to that of ZNF582-AS1.”

  1. Line 322, please state the version of the reference genome.

Response:

The reference genome version has been included as chr19:56,393,312-56,414,800 (GRCh38/hg38)

  1. Line 322-325, please list some reference papers to support the statement.

Response:

The references listed below have been added to the manuscript as reference 50-52. These literatures support our speculation that transcription factors with binding motifs encompassing the transcriptional start site (TSS) tend to have a repressing effect on gene transcription.

  1. Kannan Tharakaraman, Olivier Bodenreider, David Landsman, John L. Spouge, Leonardo Mariño-Ramírez, The biological function of some human transcription factor binding motifs varies with position relative to the transcription start site, Nucleic Acids Research, Volume 36, Issue 8, 1 May 2008, Pages 2777–2786, https://doi.org/10.1093/nar/gkn137
  2. Tabach Y, Brosh R, Buganim Y, Reiner A, Zuk O, Yitzhaky A, et al. (2007) Wide-Scale Analysis of Human Functional Transcription Factor Binding Reveals a Strong Bias towards the Transcription Start Site. PLoS ONE 2(8): e807. https://doi.org/10.1371/journal.pone.0000807
  3. Whitfield, T.W., Wang, J., Collins, P.J. et al. Functional analysis of transcription factor binding sites in human promoters. Genome Biol 13, R50 (2012). https://doi.org/10.1186/gb-2012-13-9-r50
